# Improving seasonal predictions of German Bight storm activity

Daniel Krieger[1], Sebastian Brune[1], Johanna Baehr[1], and Ralf Weisse[2]

[1]Institute of Oceanography, Universität Hamburg, Hamburg, Germany
[2]Institute of Coastal Systems – Analysis and Modeling, Helmholtz-Zentrum Hereon, Geesthacht, Germany

**Correspondence:** Daniel Krieger (daniel.krieger@uni-hamburg.de)

**Abstract**

Extratropical storms are one of the major coastal hazards along the coastline of the German Bight, the southeastern part of the North Sea, and a major driver of coastal protection efforts. However, the predictability of these regional extreme events on a seasonal scale is still limited. We therefore improve the seasonal prediction skill of the Max-Planck-Institute Earth System
Model (MPI-ESM) large-ensemble decadal hindcast system for German Bight storm activity (GBSA) in winter. We define GBSA as the 95th percentiles of three-hourly geostrophic wind speeds in winter, which we derive from mean sea-level pressure (MSLP) data. The hindcast system consists of an ensemble of 64 members, which are initialized annually in November and cover the winters of 1960/61–2017/18. We consider both deterministic and probabilistic predictions of GBSA, for both of which the full ensemble produces poor predictions in the first winter. To improve the skill, we observe the state of two physical
predictors of GBSA, namely 70 hPa temperature anomalies in September, as well as 500 hPa geopotential height anomalies in November, in areas where these two predictors are correlated with winter GBSA. We translate the state of these predictors into a first guess of GBSA and remove ensemble members with a GBSA prediction too far away from this first guess. The resulting subselected ensemble exhibits a significantly improved skill in both deterministic and probabilistic predictions of winter GBSA. We also show how this skill increase is associated with better predictability of large-scale atmospheric patterns.

## 1 Introduction

The coastline of the German Bight, which is shared by the neighboring countries of Germany, Denmark, and the Netherlands, is frequently affected by strong extratropical cyclones and their accompanying hazards, such as storm surges. These extreme events repeatedly issue challenges to coastal protection agencies, emergency management, and other interests in the region. Therefore, local actors and stakeholders may benefit from skillful predictions of these events on a seasonal-to-decadal scale.

Still, skillful predictions of storm activity on a regional scale are a challenging task, even with today's state-of-the-art modeling capabilities.

In the research field of seasonal predictions, considerable progress has been achieved over the course of the past decade. Several studies have demonstrated that current climate models show prediction skill for many large-scale atmospheric modes of the Earth system on time scales that go beyond the confines of conventional weather forecasting, as for example for the

northern hemisphere winter climate (e.g., Fereday et al., 2012), the winter North Atlantic Oscillation (NAO; Athanasiadis et al., 2014; Scaife et al., 2014a; Dunstone et al., 2016; Athanasiadis et al., 2017) and its link to the stratosphere (Scaife et al., 2016), and to some extent the Arctic Oscillation (AO; Riddle et al., 2013; Kang et al., 2014). Further studies were able to show how this good representation of large-scale atmospheric drivers in seasonal prediction systems could be used to predict climate extremes such as windstorms in the northern extratropics (e.g., Renggli et al., 2011; Befort et al., 2018; Hansen et al., 2019; Degenhardt et al., 2022).

On a more local scale, a recent study focussing on the predictability of German Bight storm activity (GBSA) has indicated that, with a carefully chosen approach, a large model ensemble, and an evaluation of different forecast categories, probabilistic predictions of high storm activity can be skillful for averaging periods longer than 5 years (Krieger et al., 2022). Krieger et al. (2022) also showed, however, that the predictive skill for single lead years in general and the next year in particular is often low and barely statistically significant, even when using a large-ensemble decadal prediction system. Using the Met Office Global Seasonal forecast System 5 (GloSea5), Scaife et al. (2014a) found large areas of positive skill for winter storminess over the North Atlantic regions, but only non-significant correlations of 0.15-0.3 over the German Bight. Degenhardt et al. (2022) also found that, using a newer version of GloSea5, even though several storm metrics are skillfully predictable over large parts of the Northeast Atlantic Ocean, the skill for the German Bight is somewhat lower than in adjacent regions. While Krieger et al. (2022) did not explicitly investigate the predictability of GBSA on a seasonal scale, the low skill for lead year 1 warrants an investigation into the seasonal predictability of GBSA and its potential for improvement.

Even before the onset of advanced computational numerical modelling, methods were developed to advance the predictability of the climate system. Lorenz (1969) proposed the idea of analogue forecasting, a prediction method which builds on the hypothesis that two observed states of the atmosphere which closely resemble each other but are temporally disconnected (*analogues*) evolve in a similar manner. As the amount of available observations, reanalyses, and climate model experiments has grown significantly over the last few decades, more data have become available that foster climate reconstruction and predictions attempts through this method (e.g., Van den Dool, 1994; Schenk and Zorita, 2012; Delle Monache et al., 2013; Menary et al., 2021). Closely related to analogue forecasting, another method has recently emerged which uses observable physical predictors of climate phenomena to estimate the future state of these phenomena. Previous studies using this technique have demonstrated that, on seasonal timescales, predictions for the state of large-scale modes of atmospheric variability like the NAO can be improved through the use of known atmospheric and oceanic teleconnections (e.g., Dobrynin et al., 2018). These studies used first-guess predictions based on the state of multiple physical predictors to refine large model ensembles and thereby reduce model spread. Similar ensemble subselection techniques have also been used to increase the predictability of the European summer climate (Neddermann et al., 2019) and European winter temperatures (Dalelane et al., 2020). This predictor-based ensemble subselection method, however, has not been applied to small-scale climate extremes like storm activity yet.

The storm climate of west-central Europe, and in particular the German Bight, is subject to a prominent multidecadal variability (e.g., Krueger et al., 2019; Krieger et al., 2021), which is arguably responsible for the comparably high predictability of GBSA a decadal scale, especially for multi-year averages (Krieger et al., 2022). Additionally, GBSA is connected to the

large-scale atmospheric circulation in the Northern Hemisphere. GBSA has shown to correlate positively with the NAO, however the strength of this connection is subject to large fluctuations on a multidecadal scale. Other atmospheric phenomena during the winter season, such as the widely studied sudden stratospheric warmings, also play a role for the extratropical storm climate, since they influence the tropospheric weather regimes (e.g., Baldwin and Dunkerton, 2001; Song and Robinson, 2004; Domeisen et al., 2013, 2015) and are able to suppress or shift surface weather patterns in the mid-latitudes, sometimes even in a way that is contrary to the state of the NAO (Domeisen et al., 2020).

Peings (2019) found that a blocking pattern over the Ural region in November can be used to identify an increased likelihood of stratospheric warmings in the subsequent winter, which in turn favor blocking setups and thus lower-than-usual storm activity over west-central Europe. Siew et al. (2020) confirmed this connection to be part of a troposphere-stratosphere causal link chain with a typical timescale of 2–3 months. The results of Peings (2019) and Siew et al. (2020) suggest that the status of the Rossby wave pattern in November might be usable as a predictor for the German Bight storm climate in the subsequent winter season.

The state of the stratospheric polar vortex in winter has also been linked to the Quasi-Biennial Oscillation (QBO) via the Holton-Tan effect (e.g., Ebdon, 1975; Holton and Tan, 1980). The Holton-Tan effect proposes a connection between easterly QBO phases, which are characterized by easterly wind and negative temperature anomalies in the lower stratosphere, and a weakened stratospheric polar vortex and thus positive stratospheric temperature anomalies in the polar Northern Hemisphere. The mechanism behind this effect has been widely studied and confirmed, e.g., by Lu et al. (2014). While some studies have already looked into the simultaneous occurrence of QBO anomalies and shifts in the European winter climate and associated windows of opportunity for better predictability (e.g., Boer and Hamilton, 2008; Marshall and Scaife, 2009; Scaife et al., 2014b; Wang et al., 2018), the state of the tropical stratosphere has not been used as a predictor for the upcoming winter storm climate in west-central Europe yet.

In this paper, we thus show that the predictability of German Bight storm activity on a seasonal scale is inherently low, but can be significantly improved through the combined use of tropospheric and stratospheric physical predictors. Drawing on the proposed links between the European winter storm climate and the Rossby wave pattern, as well as the state of the tropical stratosphere, we use temperature anomalies in the lower tropical stratosphere in September, as well as extratropical geopotential height anomalies in the middle troposphere in November as predictors for GBSA. We generate first guesses of GBSA from these predictors and select members from our ensemble based on their proximity to the first guesses. From the large-ensemble prediction system with 64 members we generate both deterministic and probabilistic predictions of winter GBSA, both for the full and the subselected ensemble, and analyze the improvement of GBSA predictability through the subselection process. We demonstrate how, compared to the low prediction skill of the full ensemble, the subselection technique significantly increases the prediction skill. The large size of the ensemble also enables a thorough sensitivity analysis of the dependency of the skill on the subselection size.

## 2 Methods and data

### 2.1 Storm activity observations

As an observational reference for storm activity in the German Bight, we make use of the time series of winter GBSA from Krieger et al. (2021). The GBSA proxy in Krieger et al. (2021) is defined as the standardized 95th seasonal (December–February, DJF) percentiles of geostrophic winds. These geostrophic wind speeds were originally calculated from three-hourly observations of mean sea-level pressure (MSLP) along the German Bight coast in Denmark, Germany, and the Netherlands, and cover the period of 1897/98–2017/18.

### 2.2 MPI-ESM-LR decadal hindcasts

In this study, we employ the extended large-ensemble decadal hindcast system based on the Max Planck Institute Earth System Model (MPI-ESM) in low-resolution (LR) mode (Mauritsen et al., 2019; Hövel et al., 2022; Krieger et al., 2022). Even though this study focuses on the seasonal timescale, we choose decadal hindcasts over any seasonal prediction systems, as the already available MPI-ESM decadal hindcast system provides us with a large ensemble size. While the ensemble consists of 80 members in total, we base our analysis on those 64 members for which three-hourly output is available (see Krieger et al. (2022) for details). At the time of this study, we are not aware of any single-model seasonal prediction system of this ensemble size and with three-hourly MSLP output available.

The MPI-ESM is a coupled climate model with individual components for the atmosphere (ECHAM6; Stevens et al., 2013), ocean and sea ice (MPI-OM; Jungclaus et al., 2013), land surface (JSBACH; Reick et al., 2013; Schneck et al., 2013), and ocean biogeochemistry (HAMOCC; Ilyina et al., 2013). Here, we only use the atmospheric output from the ECHAM6 component, which provides us with data at a temporal resolution of three hours, a horizontal resolution of 1.875°, as well as a vertical resolution of 47 levels between 0.1 hPa and the surface (Stevens et al., 2013). The hindcasts are initialized every 1$^{st}$ November from a 16-member assimilation run, starting in 1960. We use all hindcast runs initialized between 1960 and 2017 as the observational reference time series of winter GBSA ends in 2017/18.

### 2.3 German Bight storm activity (GBSA)

To quantify storm activity in this study, we draw on an established metric that uses the statistics of the hypothetical near-surface geostrophic wind speed which is obtained from horizonal gradients of MSLP (Schmidt and von Storch, 1993). Contrary to direct wind speed observations, which often show strong inhomogeneities, long MSLP records are usually more homogeneous and therefore better suited to provide information about the long-term storm climate (e.g., Alexandersson et al., 1998; Krueger and von Storch, 2011). Since winter GBSA is not directly available as an output variable of the hindcast system, we derive it from the three-hourly MSLP output (Krieger and Brune, 2022a). We calculate winter GBSA as the standardized seasonal (December–February) 95th percentiles of three-hourly geostrophic winds over the German Bight. For every ensemble member, we individually convert the horizontal differences of MSLP at three stations in the German Bight to geostrophic wind speeds at

every time step. We then derive the 95th percentiles for every winter season and standardize them by subtracting the mean and dividing by the standard deviation of the winters 1960/61-2017/18 of the respective ensemble member. The calculation follows

the methodology of Krieger et al. (2022), however it uses seasonal instead of annual 95th percentiles. Doing so, we ensure that the calculation of GBSA in the hindcast is consistent with the derivation of observed GBSA in Krieger et al. (2021). We perform the GBSA calculations individually for every member of the hindcast.

## 2.4 Predictors of GBSA

In this study, we aim to increase the predictability of winter GBSA by refining a large ensemble by selecting individual

members that are closest to a first-guess prediction of winter GBSA. To achieve this, we first need to define predictors and the generation of first guesses.

We use fields of linearly detrended September $70\,\text{hPa}$ temperature ($T_{70}$) and November $500\,\text{hPa}$ geopotential height ($Z_{500}$) anomalies as our predictors for GBSA. The choice of the respective vertical levels ($70\,\text{hPa}$ for temperatures and $500\,\text{hPa}$ for geopotential height), as well as the choice of September for $T_{70}$ is based on a lead-lag correlation analysis between winter

GBSA and temperature, as well as geopotential height fields at different levels and lead times. From this correlation analysis, September $T_{70}$ and November $Z_{500}$ emerged as the best-fitting combinations of lead time and vertical level (not shown). The data for these predictor fields are taken from the ERA5 reanalysis (Hersbach et al., 2020), which in its current state dates back to the year 1940. Anomalies are calculated by subtracting the 1940–2017 mean from the time series. We ensure that there are regions where the correlation coefficient between the predictor and GBSA is significantly different from zero over the whole

investigation period (1940–2017 for predictors, winters 1940/41–2017/18 for GBSA).

In every prediction year, we generate a first guess of winter GBSA from the state of our chosen predictors. For each predictor $x_p$, we first analyze which gridpoints show a locally significant positive correlation with GBSA for all years from 1940 to the year before the initialization ($p \leq 0.05$). The statistical significance of the correlation is determined through a gridpoint-wise 1000-fold bootstrapping with replacement (Kunsch, 1989; Liu and Singh, 1992), where the 0.025 and 0.975 quantiles of

145 bootstrapped correlations define the range of the 95 % confidence interval. If the 95 % confidence interval excludes a value of $r = 0$, we consider the correlation for this gridpoint significant and that gridpoint is taken into account for the generation of a first guess. As both the anomalies of the predictors and the index of winter GBSA are defined as standardized anomalies following a Gaussian normal distribution with a mean of 0 and a standard deviation of 1, we can directly translate the state of each predictor into a first guess of our predictand GBSA. Therefore, we compute the first guess of the predictand (GBSA) as

an area-weighted average $y_p$ of the state of the predictor $x_p$ for those gridpoints $(i,j)$ that show a locally significant positive correlation with GBSA, following Eq. (1).

$$y_p = \frac{\sum\limits_{i=1,j=1}^{sig} x_p(i,j)\cos\Phi_j}{\sum\limits_{i=1,j=1}^{sig} \cos\Phi_j}. \tag{1}$$

In Eq. (1), $\cos \Phi_j$ denotes the cosine of the latitude of each gridpoint used as a weighting factor. For geopotential height anomalies, we constrain the region that can contribute to the first guess to the boreal extratropics between 30°N-90°N, as the pattern of geopotential height in this region describes the Rossby wave train which strongly governs the extratropical winter storm climate. We make sure that each predictor always contributes significantly positively correlated gridpoints in every prediction year, as the correlation strength and location of the significant correlations may vary from year to year.

In addition to the gridpoint-wise significance test, we also test the fields of $T_{70}$ and $Z_{500}$ for global significance by controlling for the false detection rate (FDR; Wilks, 2006). We achieve this by ranking the $p$-values of all $n$ gridpoints from smallest ($p_{(1)}$) to largest ($p_{(n)}$), so that

$$p_{(1)} \leq p_{(2)} \leq \cdots \leq p_{(n-1)} \leq p_{(n)}. \tag{2}$$

Subsequently, we then individually test each $p$-value against a threshold that is comprised of a predefined criterion of $\alpha_{\mathrm{FDR}} = 0.05$, scaled by the rank $i$ of the respective $p$-value and the total grid size $n$. Should a $p$-value satisfy the condition

$$p_{(i)} < \alpha_{\mathrm{FDR}} * \frac{i}{n} \tag{3}$$

we consider the correlation at this point to be significant globally. Doing so, we are able to determine whether certain regions of our predictor fields show up as locally significant only due to spatial autocorrelation of the respective atmospheric fields. Please note that for the calculation of the predictor states, we still use information from all locally significant gridpoints, regardless of whether the respective gridpoints are globally significant or not.

For every model run, we choose a number $n$ of ensemble members in our forecast ensemble with a GBSA closest to the state of the two predictors $T_{70}$ and $Z_{500}$ in that respective year. Closeness is hereby defined as the absolute difference between the predicted GBSA of the respective member and the state of the predictor. Because we select $n$ members twice in every run, i.e., once for every predictor, and the two selections of members might overlap, the size of this resulting subselection can vary between $n$ members – if the states of $T_{70}$ and $Z_{500}$ are identical in that year – and $2n$ members – if the states of $T_{70}$ and $Z_{500}$ are far enough apart that there is no overlap between the selected members. From this resulting subselection, we then calculate deterministic and probabilistic GBSA predictions. A schematic overview of the predictor-based subselection is given in Fig. 1. Deterministic predictions are computed by averaging the GBSA predictions over all members in the subselection. For probabilistic predictions, we calculate the fraction of members within the subselection that exceed a defined threshold for high storm activity of 1 standard deviation above the long-term mean. It should be noted that selected members are weighted equally in all computations, even though some of them might have been selected by both predictors.

## 2.5 Skill metrics

To evaluate the improvement of prediction skill for winter GBSA, we first define separate skill metrics for deterministic and probabilistic model predictions.

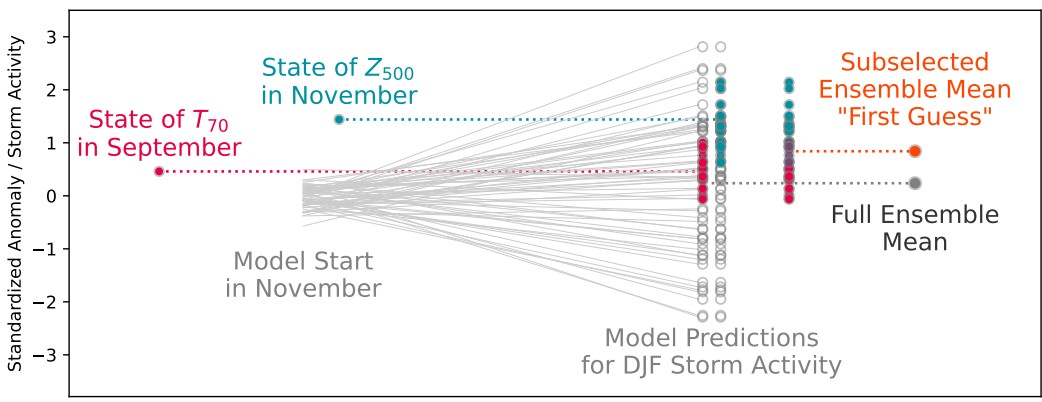

**Figure 1.** Schematic depiction of the predictor-based subselection workflow, adapted from Dobrynin et al. (2018).

We measure the skill of deterministic predictions with Pearson's anomaly correlation coefficient (ACC) and the root-mean-square error (RMSE) between predicted and observed quantities. The ACC is defined as

$$\mathrm{ACC} = \frac{\sum_{i=1}^{N}(f_i - \bar{f})(o_i - \bar{o})}{\sqrt{\sum_{i=1}^{N}(f_i - \bar{f})^2 \sum_{i=1}^{N}(o_i - \bar{o})^2}}, \tag{4}$$

where $f_i$ and $o_i$ denote predictions and observations at a time step $i$, and $\bar{f}$ and $\bar{o}$ mark the long-term averages of predictions and observations. ACC values of 1 indicate a perfect correlation, 0 no correlation, and -1 a perfect anticorrelation. The statistical significance of the ACC is again determined through a 1000-fold bootstrapping with replacement and a significance criterion of $p \leq 0.05$.

The RMSE is calculated from the predicted and observed quantities $f_i$ and $o_i$ by

$$\mathrm{RMSE} = \sqrt{\frac{1}{N}\sum_{i=1}^{N}(f_i - o_i)^2}. \tag{5}$$

Probabilistic predictions of high storm activity are tested against a climatology-based reference prediction and evaluated with the strictly proper Brier skill score (BSS; Brier, 1950). The climatology-based reference prediction is constructed from the climatological frequencies of observed GBSA (e.g., Wilks, 2011). Here, we draw on the definition of GBSA from Krieger et al. (2021) which assumes an underlying Gaussian normal distribution.

We calculate the BSS as follows:

$$\mathrm{BSS} = 1 - \frac{\mathrm{BS}}{\mathrm{BS_{cli}}}. \tag{6}$$

BS and $\mathrm{BS_{cli}}$ indicate the Brier Scores of the probabilistic model prediction and the fixed climatological reference prediction, respectively. Positive values show that the model predictions perform better than the climatology-based predictions and vice versa. A BSS of 1 would indicate a perfect model prediction, i.e., all members of the ensemble predicting the occurrence or absence of a high-storm-activity event correctly in every year.

The individual Brier Scores BS are defined as

$$\text{BS} = \frac{1}{N} \sum_{i=1}^{N} (F_i - O_i)^2, \tag{7}$$

where $F_i$ and $O_i$ denote predictions and observations at a time step $i$. In the model, we calculate the predicted probability $F_i$ from the fraction of ensemble members that predict a high-storm-activity event. For the climatology-based prediction, $F_i$ is a fixed value. As high storm activity is defined via a threshold of one standard deviation above the mean state, we calculate the climatological probability of a high-storm-activity event occuring to be $F_i = 1 - \Phi(1) = 0.1587$, where $\Phi(x)$ is the cumulative distribution function of the Gaussian normal distribution. This means that the probability of a random sample from a Gaussian normal distribution with a mean of $\mu$ and a standard deviation of $\sigma$ being larger than $\mu + 1\sigma$ is slightly less than $16\%$. The observed probability $O_i$ always takes on a value of either 1 or 0, depending on whether the event happened or not.

## 2.6 Training and hindcast periods

The recent backward extension of the ERA5 reanalysis extends the dataset back to 1940. Because the predictions of GBSA are based on predictors that are derived from regions where the predictor and GBSA correlate significantly, we require a sufficiently long training period to identify these regions before the start of the first model run. Hence, we classify the first two decades (1940–1959), for which only ERA5 and observational GBSA data are available, as the training-only period, and start the actual predictor-based first guesses of GBSA in the year 1960. Doing so, we can ensure that we only use reanalysis data to predict GBSA that was already available at the start of the respective hindcast run, but still use the full range of hindcasts which begin in 1960. The hindcast period, i.e., the period in which we predict GBSA and assess the skill of the model and the subselection, is thus confined to a total of 58 winters from 1960/61 to 2017/18.

## 2.7 Composites

To check whether our prediction mechanism is also physically represented in the hindcast, we calculate composites of $T_{70}$ and $Z_{500}$ in the years with highest and lowest modeled DJF GBSA, respectively. We use all initialization years (1960–2019), all 64 members, and all lead years except the first one after the initialization (2–10), leaving us with 34560 model years. From these 34560 years, we select the 100 highest and lowest GBSA winters, compute composite mean fields of both predictors in the respective years preceding these winters, and calculate the difference between the composites of high and low GBSA. We then analyze the patterns of the composite differences to determine whether they resemble the correlation patterns between the predictors in ERA5 and observed DJF GBSA.

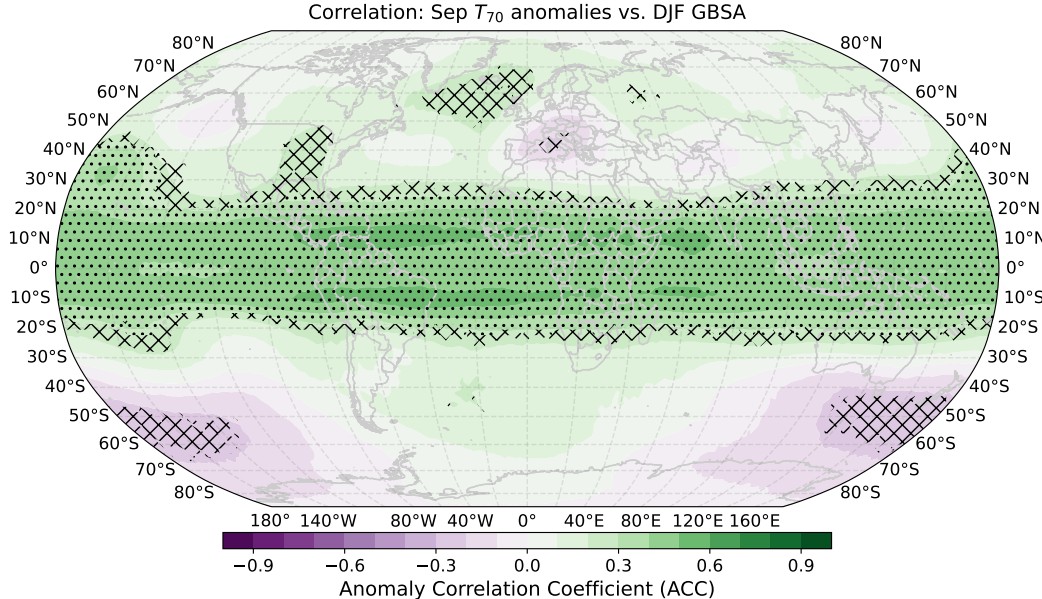

**Figure 2.** Gridpoint-wise correlation coefficients between global $T_{70}$ anomalies in ERA5 and observed winter (DJF) German Bight storm activity. Period 1940–2017 for temperature anomalies, 1940/41–2017/18 for storm activity. Hatching indicates local statistical significance ($p \leq 0.05$) determined through 1000-fold bootstrapping. Stippling indicates additional global field significance by controlling for the FDR at a level of $\alpha_{\mathrm{FDR}} = 0.05$.

## 3 Results

### 3.1 Correlations of predictor fields with winter storm activity

We identify $T_{70}$ and $Z_{500}$ anomalies as physical predictors for winter GBSA. To illustrate the connection between the global fields of these two predictors and storm activity, and to demonstrate which regions mainly contribute to the first-guess predictions, we correlate gridpoint-wise time series of $T_{70}$ and $Z_{500}$ anomalies with observed winter GBSA for the entire time period of 1940–2017.

     The highest correlations between GBSA and $T_{70}$ anomalies are found in the tropics in a circumglobal band between roughly

15°N and 15°S, with values as high as 0.5–0.6 (Fig. 2). Notably, correlations are slightly lower directly at the equator than a few degrees north and south of it. Over Europe, a smaller region with slightly negative correlations is present, surrounded by slightly positive correlations to the northeast and northwest. Over the Southern Ocean, a signal of slightly negative correlations emerges as well. However, none of the regions outside of the tropics correlate with DJF GBSA as high as the tropics themselves. In total, 21.1% of all locally significant gridpoints (or 6.8% of all gridpoints) fail the global field significance test, indicating

random correlation. Most of these gridpoints belong to regions outside the tropics, which reinforces the hypothesis that the tropical stratospheric temperatures show the strongest connection to winter GBSA.

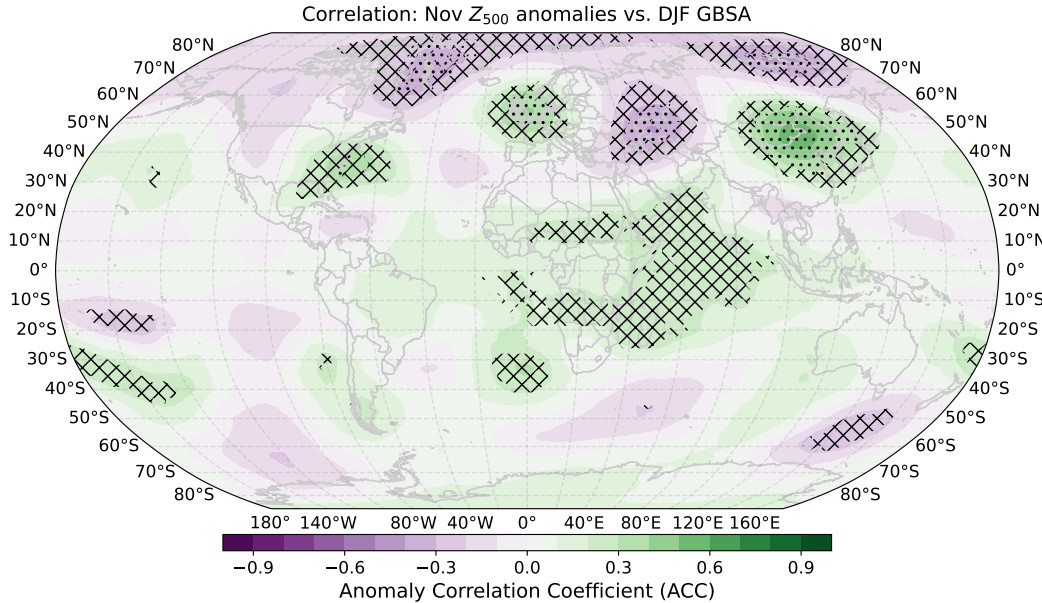

**Figure 3.** Gridpoint-wise correlation coefficients between global $Z_{500}$ anomalies in ERA5 and observed winter (DJF) German Bight storm activity. Period 1940–2017 for geopotential height anomalies, 1940/41–2017/18 for storm activity. Hatching indicates local statistical significance ($p \leq 0.05$) determined through 1000-fold bootstrapping. Stippling indicates additional global field significance by controlling for the FDR at a level of $\alpha_{FDR} = 0.05$.

For $Z_{500}$ anomalies, the strongest positive correlations with winter GBSA are found over the British Isles and the adjacent Northeast Atlantic, as well as over East-central Asia and the North American East Coast with peaks around 0.4 (Fig. 3). The strongest negative correlations emerge over East-central Europe, Greenland, and northeastern Siberia, reaching as low as -0.4.

The correlation pattern in the boreal extratropics is in line with the findings of Peings (2019) and Siew et al. (2020) in a way that troughing (i.e., the opposite of ridging) over the Ural region and thus a reduced likelihood of stratospheric warmings in the following winter season is connected to higher-than-usual storm activity in the German Bight. These areas of significant correlations also strongly resemble a Rossby wave pattern which spans the boreal extratropics. Across the subtropical and tropical latitudes, some areas of slightly positive correlations can be found over the Sahel region and the Indian Ocean. Together

with the negative correlations in the Arctic, these significant areas may be indicative of a relation to the Arctic Oscillation (AO; Thompson and Wallace, 1998). In the Southern Hemisphere, small patches of slightly positive and negative correlations are distributed circumglobally. However, the absolute correlations of the aforementioned regions in the tropics and the Southern Hemisphere are much lower than those in the northern extratropics. In total, 81.8% of locally significant gridpoints (or 13.1% of all gridpoints) fail the global field significance test, leaving just the the regions associated with the Rossby wave pattern

globally significant. This test supports the decision of only taking the boreal extratropics into account for the calculation of the $Z_{500}$ predictor states.

## 3.2 Improvement of GBSA predictability

We use the established connection between $T_{70}$ and $Z_{500}$ anomalies and DJF German Bight storm activity to predict the storm activity of the upcoming winter season for the hindcast period of 1960–2017. We use latitude-weighted field means of $T_{70}$ and $Z_{500}$ in ERA5 as our initial guess for DJF storm activity. Since both the time series of temperature and geopotential height anomalies and those of GBSA are standardized, we do not need to apply a scaling factor to translate the field means of temperature and geopotential height anomalies to GBSA. We only use information from data between 1940 and the year of the start of the forecast. Thus, the amount and distribution of gridpoints that are included in the calculation of the first-guess prediction can vary from year to year. To generate first-guess predictions of winter GBSA, we need to select a certain number of ensemble members closest to the initial guess for each predictor.

One degree of freedom in this process is the sampling size, i.e., the number of members selected for each predictor. The choice of this sampling size has an effect on the skill metrics of the subselected ensemble predictions. To illustrate the dependency of the model skill on the sample size, we test the correlation, RMSE, and high-activity BSS against climatology for all sample sizes between 1 and 64 (Fig. 4a) for the hindcast period of 1960–2017. Furthermore, we perform these sensitivity studies for both predictors individually to show how the combined use of both predictors changes the skill compared to just using one of the two (Fig. 4b and 4c).

The sensitivity analysis for the combined use of both predictors (Fig. 4a) shows a strong increase in correlation to above 0.6 for up to roughly 50 members. This indicates that removing only about one sixth of all members per predictor is sufficient to increase the correlation between the deterministic prediction and observations significantly. The optimal sample size for correlations is found at 25 members per predictor ($r = 0.64$). For the RMSE, smaller sample sizes between 10 and 40 members yield the biggest improvement, with an optimum at 25 members ($\mathrm{RMSE} = 0.70$). The BSS can be maximized by selecting 25 members for each predictor as well ($\mathrm{BSS} = 0.28$), and shows a similar window of opportunity as the RMSE between 10 and 40 members. The sensitivity analysis for $T_{70}$ alone (Fig. 4b) reveals a slightly lower potential for probabilistic skill improvements. Here, the BSS can be increased to 0.23 with a sample size of 33 members, but a deterioration of the BSS compared to the full ensemble occurs below 10 members. Similarly, choosing $Z_{500}$ alone (Fig. 4c) only improves probabilistic forecasts when selecting more than 25 members with a maximum of 0.16 at 45 members.

The deterministic skill metrics also show similar windows of opportunity for both predictors individually. While correlation and RMSE for $Z_{500}$ are maximized at sample sizes of 44 members ($r = 0.5$, $\mathrm{RMSE} = 0.79$) the optimum for $T_{70}$ is located at 42 members ($r = 0.55$, $\mathrm{RMSE} = 0.77$). It should be noted that, for both predictors, the optimal sample sizes for RMSE and correlation are equal, since the correlation coefficient and RMSE are directly related for standardized sets of forecasts and observations (Barnston, 1992). Just like for the BSS, the individual contributions of the predictors to correlation and RMSE are smaller than the combined effect, manifesting the need to combine multiple predictors in the subselection to achieve the best possible skill increase.

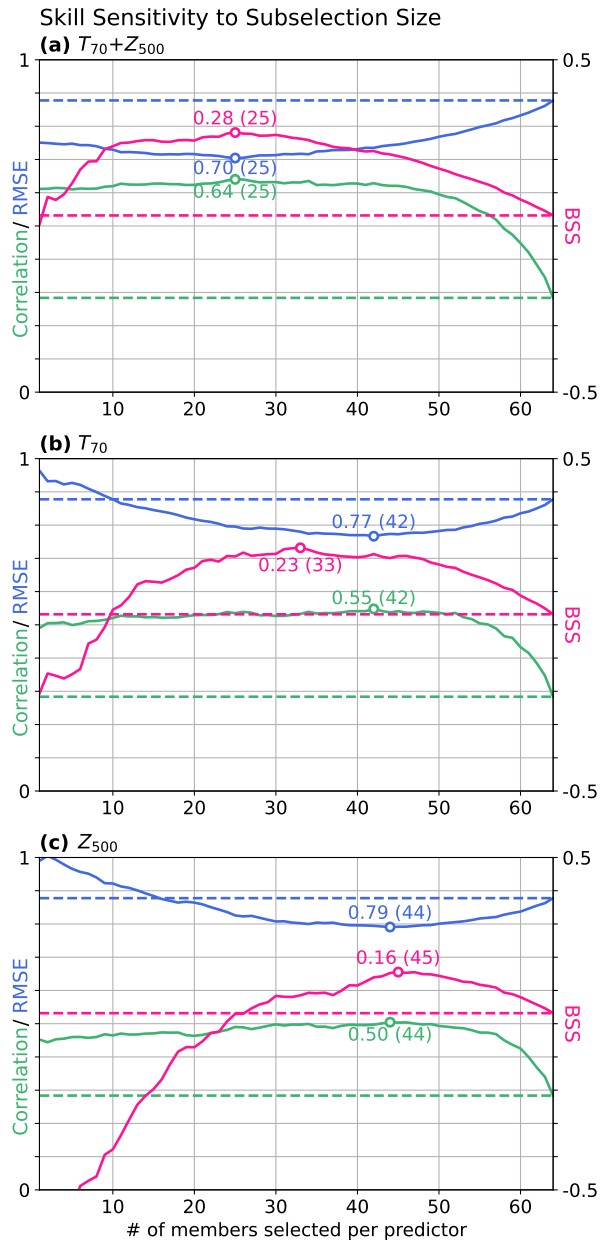

**Figure 4.** Dependency of various skill scores (ACC (green), RMSE (blue), and BSS for high storm activity against climatology (pink)) of model ensemble predictions of DJF GBSA on the sample size chosen for each predictor during the subselection. The subselection is performed based on **(a)** both predictors, **(b)** only $T_{70}$, and **(c)** only $Z_{500}$. Dashed baselines show the respective skill scores of the full 64-member ensemble. Optimal skill scores (highest ACC and BSS, lowest RMSE) are displayed as annotated dots, together with the optimal sample size in brackets.

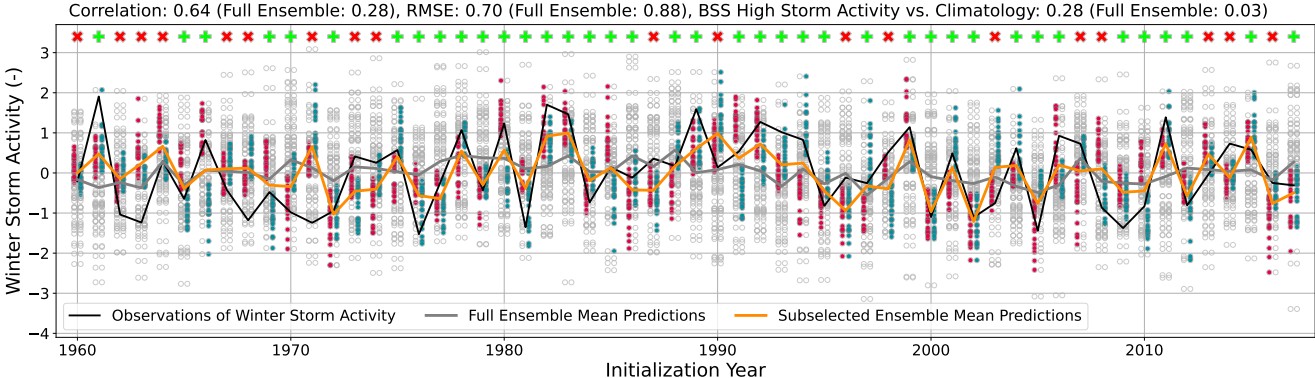

**Figure 5.** Predictions of DJF GBSA by the 64-member ensemble mean (gray line), the subselected ensemble mean (orange line), as well as observed DJF GBSA (black line). Period 1960–2017 for model initializations, 1960/61–2017/18 for storm activity observations. Circles indicate GBSA predictions of individual members, colored circles indicate the selected 25 members closest to the first-guess predictions based on $T_{70}$ (red), and $Z_{500}$ (teal). Green plus signs and red "x" markers denote forecasts where the subselection is closer to or further away from the observation than the full ensemble.

From the sensitivity study, we find that sample sizes of 20–30 members constitute a fair compromise between the optimal sample sizes of deterministic and probabilistic predictions. Therefore, we exemplarily analyze the prediction of winter GBSA in the hindcast period for a subselection size of 25 members per predictor in greater detail (Fig. 5).

Over the forecast period, the first-guess estimates obtained from combining $T_{70}$ and $Z_{500}$ anomalies and observed winter GBSA correlate well (0.64), an improvement of 0.36 from the deterministic full-ensemble model prediction. The subselected ensemble captures the variability in DJF GBSA much better than the full 64-member ensemble. High agreements between first-guess predictions and observations are found in the late 1970s, the 1980s, as well as between the mid-1990s and the mid-2000s. With an RMSE of 0.70, the subselection-based prediction shows a slightly lower error than the full ensemble (0.88). Furthermore, the BSS against climatology of the reduced ensemble for high storm activity predictions is greatly increased to 0.28, compared to 0.03 for the full 64-member ensemble. In 39 out of the 58 individual predictions (67 %), the subselection leads to an improvement in the prediction as measured by the absolute difference between ensemble mean and observations.

Overall, all three metrics show a significant improvement for the first-guess-based reduced ensemble, revealing that both deterministic and probabilistic storm activity predictions can be significantly improved by the combined inclusion of $T_{70}$ and $Z_{500}$ as physical predictors.

### 3.2.1 Skill increase for large-scale atmospheric variables

In order to determine on a physical basis why the subselected ensemble shows a higher prediction skill for GBSA in both deterministic and probabilistic modes, we analyze the change in ACC between the full ensemble mean and the mean of the subselected ensemble for three atmospheric variables that can be associated with the state of the winter climate over Europe

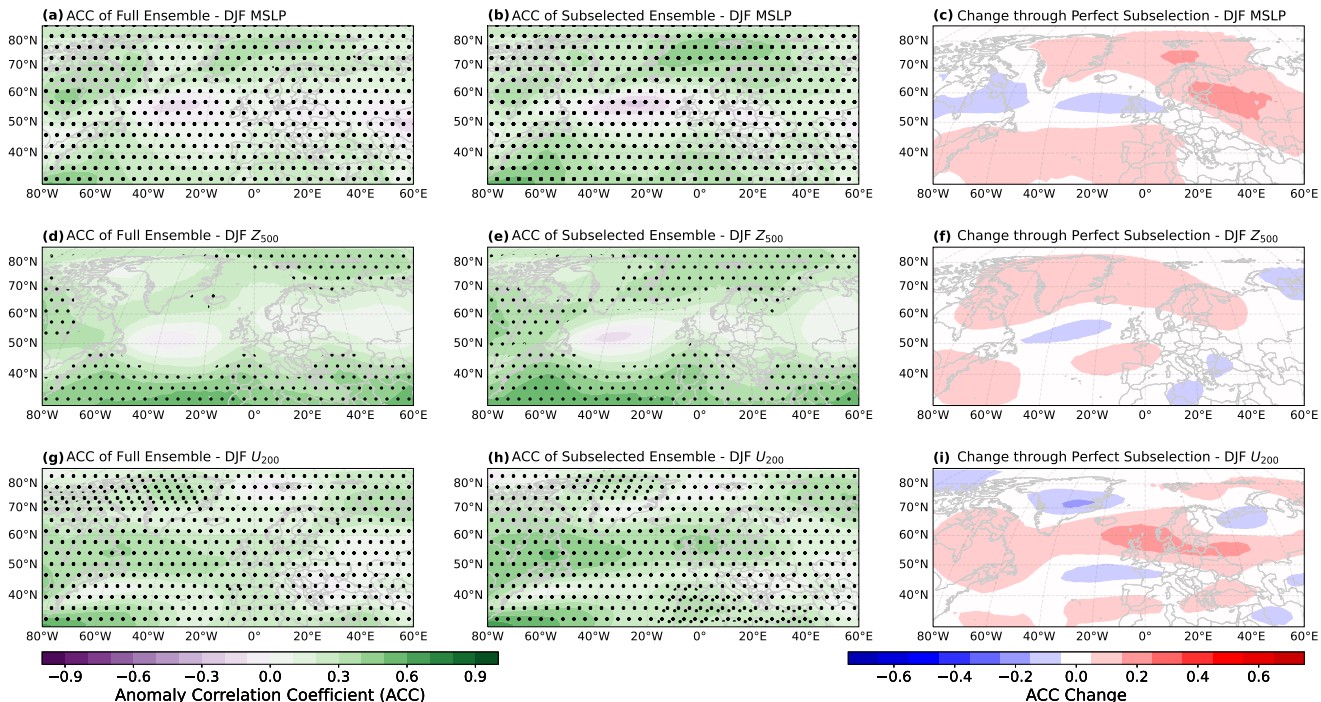

**Figure 6.** Anomaly correlation coefficients (ACC) for ensemble mean predictions of the full 64-member ensemble (left column), the 25-member subselection (middle column), and the change in ACC between the full and subselected ensemble (right column) for winter-mean (DJF) MSLP anomalies (first row), 500 hPa geopotential height anomalies ($Z_{500}$, second row), and 200 hPa zonal wind anomalies ($U_{200}$, third row). Winter-mean anomalies are calculated by averaging monthly anomalies from December, January, and February. Period 1960/61–2017/18. Stippling indicates statistical significance ($p \leq 0.05$) determined through 1000-fold bootstrapping.

(Fig. 6). We choose one variable that we also use for the ensemble subselection, winter-mean 500 hPa geopotential height ($Z_{500}$), as well as two variables that are not included in the ensemble subselection, namely winter-mean MSLP, and 200 hPa zonal wind ($U_{200}$). Variations in MSLP indicate the prevalent distribution of high and low pressure areas, which directly
influence the near-surface wind speed and can be indicative of the mean wind climate during winter. The field of $Z_{500}$ provides insight into the state of the Rossby wave pattern in winter and whether the large-scale mid-tropospheric flow diverts storms away from or towards the German Bight. The location and strength of the polar jet stream, expressed as $U_{200}$, governs the lower tropospheric setup and can enhance or suppress the formation of storms.

We find that the full ensemble shows significant skill for deterministic winter MSLP forecasts north of 60°N, as well as for
winter $Z_{500}$ south of 45°N, but limited skill for both MSLP and $Z_{500}$ over west-central Europe and the adjacent region of the North Atlantic Ocean (Fig. 6a and 6d). The subselected ensemble shows a slightly higher skill for MSLP over Scandinavia and the Iberian Peninsula, but not over the German Bight and more generally west-central Europe (Fig. 6b and 6c). The skill of the subselection for $Z_{500}$ is also slightly improved from Greenland to northern Scandinavia (Fig. 6e and 6f). Despite not

showing an improvement over the German Bight, higher skill north and south of the German Bight indicates an increase in the predictability of the meridional gradient of MSLP and $Z_{500}$, which is crucial to more accurately predict the wind climate in the German Bight. For $U_{200}$, the full ensemble shows significant skill in a mostly zonally-oriented band spanning from the North Atlantic around 55°N into west-central Europe (Fig. 6g). Notably, positive correlations are located closer to the German Bight than for MSLP and $Z_{500}$. The subselected ensemble mostly retains this correlation pattern, but extends the significant skill across the German Bight into east-central Europe (Fig. 6h and 6i). The improvement in predictability of $U_{200}$, which is associated with the strength and location of the jet stream, is in accordance with the improvement in GBSA prediction skill, as the jet stream governs the formation and intensification of extratropical cyclones.

### 3.2.2 Potential capabilities of the model (perfect test)

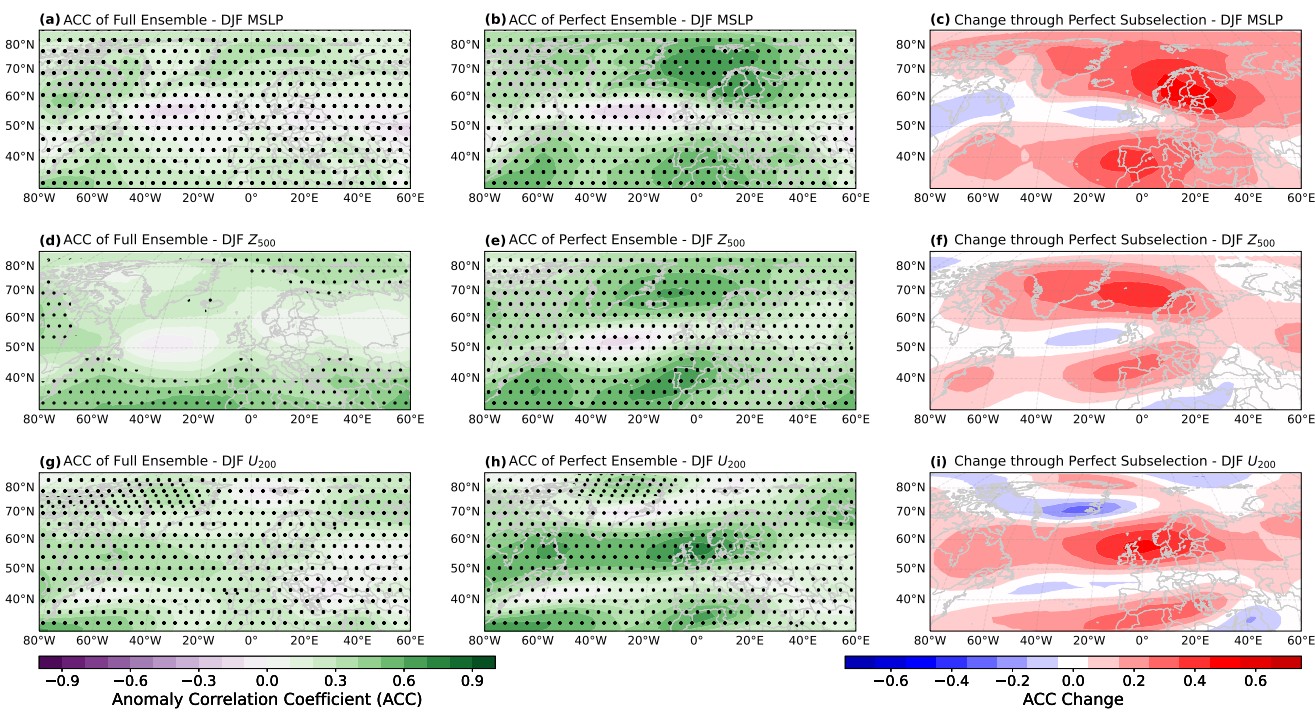

**Figure 7.** Like Fig. 6, but for a perfect test, i.e., the 25 members closest to the actually observed GBSA are selected.

To determine the theoretical maximum of skill improvement that the model could achieve, we perform a perfect test. Instead of choosing the 25 members closest to the first-guess winter GBSA determined from the two respective predictors as in the previous section, the perfect test selects those 25 members in each forecast that are closest to the actually observed winter GBSA. We refer to the set of these members selected in the perfect test as the perfect ensemble. For this perfect ensemble, we again analyze the change in ACC for MSLP, $U_{200}$, and $Z_{500}$ (Fig. 7). Note that operationally the perfect test would require information from the future, as the selection of a perfect ensemble at the start of the forecast in November relies on observational

data which is not available until the end of February of the following year. For this reason, the perfect test is merely a tool of retrospective model evaluation and can not be replicated operationally. Again, we find that the greatest skill increases occur in regions where the full ensemble already showed significant skill. For MSLP and $Z_{500}$, the skill north and south of the German Bight and therefore the predictability of the meridional gradient is significantly improved, while the skill in a region near and slightly west of the German Bight is almost unaffected by selecting the perfect ensemble (Fig. 7b, 7c, 7e, and 7f). Even with knowledge of future GBSA, the perfect ensemble is not able to significantly improve predictions of MSLP and $Z_{500}$ in the same area. The perfect ensemble also improves $U_{200}$ predictability over regions where the full ensemble already showed skill, i.e., mostly between 50°N and 65°N (Fig. 7h and 7i).

Generally, the patterns of skill increase through ensemble subselection are similar for the non-cheating hindcast and the perfect ensemble. The major difference between the two modes is that the increase in predictability of MSLP, $Z_{500}$, and $U_{200}$ is much larger in the perfect ensemble, which is to be expected as the model is able to use information from the future. From the similarity of the skill improvement patterns, however, we construe that the improvement of GBSA prediction skill through subselecting members is consistent with the physical mechanisms behind the extratropical winter storm climate and their predictability. The strong contrast in the magnitude of skill improvement points out the potential of the ensemble for even better predictions of the extratropical winter climate. However, additional research into more sophisticated ensemble refinement techniques, possibly also including the involvement of machine learning, is required to make use of this potential.

## 3.3 Representation of the mechanisms in the model

Figs. 8 and 9 show differences in composite mean modeled $T_{70}$ and $Z_{500}$ fields between years prior to modeled high- and low-storm-activity winters. The patterns of $T_{70}$ differences (Fig. 8) barely resemble the observed correlation patterns that are apparent between reanalyzed $T_{70}$ fields and DJF GBSA observations (see Fig. 2). Differences in the tropics, where observed correlations are highest, hardly exceed 0.3 K. In contrast, negative differences of up to -2 K, i.e., lower $T_{70}$ preceding high DJF GBSA, emerge in the austral extratropics, where slightly negative correlations can also be found in the observations. Overall, the model appears to be incapable of reproducing the pathway from stratospheric temperature anomalies in September to changes in the extratropical winter storm climate in the German Bight.

The patterns in the composite differences of $Z_{500}$ (Fig. 9), however, demonstrate a fair agreement with observed correlation patterns (see Fig. 3). Before high-storm-activity winters, geopotential heights in the model are up 30 gpm higher over the US East Coast, west-central Europe and northeast Asia than before low-storm-activity winters. Similarly, up to 30 gpm lower geopotential heights are modeled over Canada, Greenland, the Ural region and the Arctic in years prior to high-storm-activity winters. These regions of largest geopotential height differences match the regions of significant correlations between $Z_{500}$ in ERA5 and observed DJF GBSA. We thus conclude that the physical link between November geopotential height anomalies and subsequent DJF GBSA is very well modeled by the hindcast system.

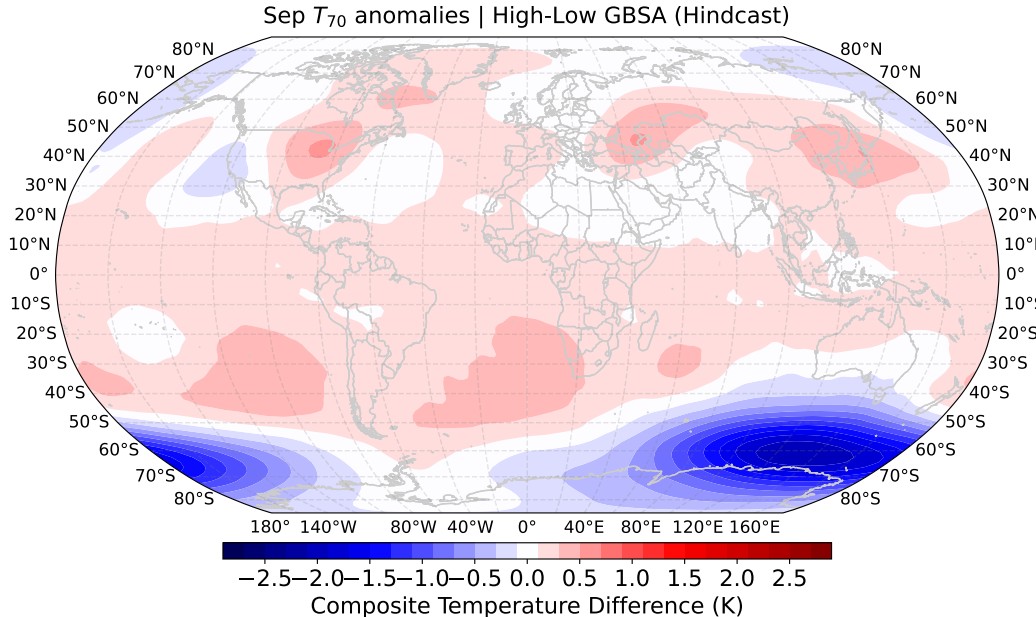

**Figure 8.** Composite mean $T_{70}$ of 100 model years with the highest subsequent DJF GBSA minus composite mean $T_{70}$ of 100 model years with the lowest subsequent DJF GBSA in MPI-ESM-LR decadal hindcast runs. Data are taken from all initializations, all members, and all lead years except for the first year after initialization.

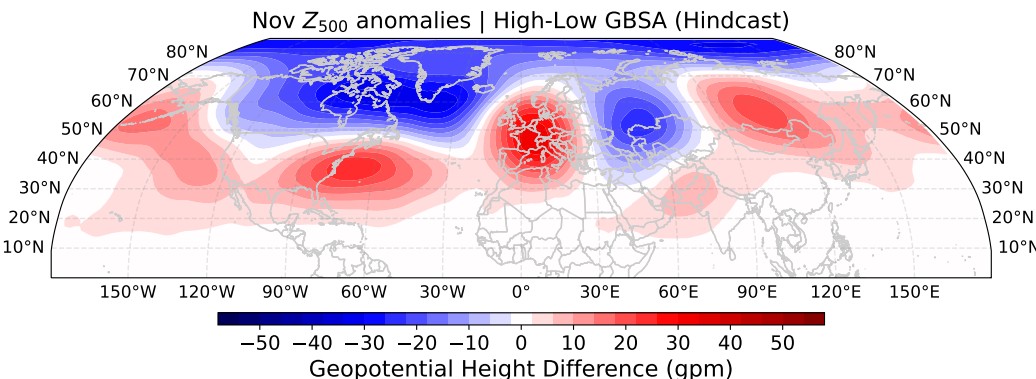

**Figure 9.** Composite mean $Z_{500}$ of 100 model years with the highest subsequent DJF GBSA minus composite mean $Z_{500}$ of 100 model years with the lowest subsequent DJF GBSA in MPI-ESM-LR decadal hindcast runs. Data are taken from all initializations, all members, and all lead years except for the first year after initialization.

## 4  Discussion

We use a decadal prediction system for seasonal predictions because we want to make use of the large ensemble size and the high temporal resolution of the model output. While a seasonal prediction system would be sufficient for this analysis, we are not aware of any available seasonal single-model initialized large ensembles with 64 members and three-hourly MSLP output. In addition, the use of the MPI-ESM-LR decadal prediction system allows us to directly compare the predictability for the first winter to the results from Krieger et al. (2022). We find that the full-ensemble prediction skill for winter GBSA ($r = 0.28$, BSS $= 0.03$) is close to what Krieger et al. (2022) found for lead-year 1 predictions of annual GBSA. This similarity is explainable by the higher wind speeds of storms in winter and thus the higher contribution of the winter season to annual storm activity than the remaining seasons.

Furthermore, the MPI-ESM-LR decadal hindcast offers a total of about 60 initialization years, while the corresponding seasonal prediction system based on the MPI-ESM-LR only covers about 40 initialization years. Additionally, the backward extension of ERA5 to 1940 allows us to define the initial training period, that is the period from which we determine areas of significant correlation between GBSA and the predictors before the first hindcast run, by two decades which fully precede the hindcast. Thus, we are able to generate predictor-based first guesses for almost six decades of hindcast initializations to test the skill of the model, while the seasonal system (in Dobrynin et al., 2018) only allowed for a hindcast period of two decades.

The improved prediction skill for winter GBSA ($r = 0.64$) is higher than what is typically achieved with full-ensemble seasonal or decadal prediction systems, especially in the German Bight, where previous studies on large-scale winter storm activity have demonstrated shortcomings of seasonal (e.g., Scaife et al., 2014a; Degenhardt et al., 2022) and decadal prediction systems (e.g., Kruschke et al., 2014, 2016; Moemken et al., 2021). This high skill for storm activity is especially impressive considering the comparably low ACC of both the full and even the subselected ensemble for winter MSLP over Europe. Here, the winter MSLP ACC values remain below ACCs of, for example, a multisystem seasonal prediction of MSLP (Athanasiadis et al., 2017) and are closer to those found by Athanasiadis et al. (2020) on the decadal scale.

While our subselection increases the skill quite notably, there is still room for more improvement. This becomes especially apparent in the perfect test plots, where the potential perfect ACC increase for associated physical parameters like $Z_{500}$, $U_{200}$, and MSLP is a lot larger compared to our predictor-based ensemble subselection. A possible method to further improve the predictability and to rely more on the model physics would be checking which members actually predicted the observed patterns in November correctly and subselect those members. However, a test which replaces the members closest to the state of the $Z500$ predictor field with those members that exhibit the highest pattern correlations with observed $Z500$ fields in November results in a smaller increase in prediction skill for GBSA (not shown). We argue that the ensemble spread in November (i.e., directly after the initialization) is too low to objectively distinguish "good" from "bad" members. This method of refining the ensemble based on the predictions of observed patterns would become more feasible if the ensemble was initialized earlier than in November or if the winter prediction was supposed to be updated during the winter, based on, for instance, the model representation of certain observed atmospheric fields in December or January.

The correlation between temperature anomalies in the tropical stratosphere and GBSA is notably higher than the correlation between the same predictor and both the wintertime North Atlantic Oscillation (NAO) and Arctic Oscillation indices, two climate modes representative for the larger-scale atmospheric circulation over the norther mid- and high latitudes. We argue that the increased correlation with GBSA is caused by the strong multidecadal signal within both the tropical stratosphere and GBSA which appears to be in phase over the investigated period. While GBSA is also connected to the NAO and the AO to a certain degree with correlations of 0.51 (NAO) and 0.40 (AO) for 1960/61–2017/18, the connection to the NAO has been shown to fluctuate over time (e.g., Krieger et al., 2021). In the 1960s, the running correlation between GBSA and the NAO index reached its minimum at values below 0.2, indicating that the decadal to multidecadal signals in both time series appear to move out of phase at times. This implies that even with an almost perfect forecast of the NAO, as for instance achieved by Dobrynin et al. (2018), a equally good prediction of GBSA cannot be guaranteed. Furthermore, we conclude that, while Scaife et al. (2014a) attribute a significant fraction of prediction skill for winter storm activity to the predictability of the NAO, our predictors may be better suited for direct GBSA predictions without simultaneously improving NAO predictions as well.

Since this is a single-model study based on the MPI-ESM-LR, our findings are model-specific. Therefore, the conclusions we draw are true for this model and the associated model physics. However, because the subselection process is purely based on the statistical relationship between reanalysis data and observations, it could also work in other large model prediction ensembles, as long as the internal variability in the ensemble encompasses the natural variability of GBSA.

We confirmed the connection between GBSA and the two chosen predictors through correlation analysis based on the ERA5 reanalysis. To ensure that the choice of reanalysis does not bias our results, we performed the correlation analysis between the predictor fields and GBSA in the NCEP-NCAR reanalysis (Kalnay et al., 1996) for the winters 1948/49–2017/18 and found similar patterns of correlations (not shown).

Despite having increased the predictability for the first winter on a seasonal scale, the decadal skill matrix for annual GBSA in Krieger et al. (2022) presents more lead times with poor predictability between lead year 1 and longer averaging periods. Using tropospheric patterns as predictors for longer lead times than the first winter is unphysical given the short memory of the troposphere. Therefore, new predictors (e.g., sea surface temperature) would need to be tested and used for an improvement of the GBSA prediction skill beyond the first winter. Alternatively, the model could be optimized to skillfully predict the state of the tropical stratosphere beyond the first year, for example via an accurate representation of the QBO. Such a prediction would then still require a statistical approach to link the QBO to GBSA, since we showed that the pathway from the tropical strato-sphere to the extratropics in the boreal winter is poorly represented in the model (Fig. 8). Looking beyond the predictability of GBSA, the ensemble subselection method may be usable to improve the predictability of other climate extremes that can be associated with physical precursors. As long as the internal variability of a prediction system is able to capture the variability of the predicted event or extreme, and precursors with a stationary link to the event are found, an improvement of the prediction skill appears feasible. This method is also not limited to a certain timescale, so that the same approach may as well be usable in decadal prediction, but also in sub-seasonal or weather prediction. Any further analysis in this direction, however, is beyond the scope of this study.

## 5 Conclusions

We showed that the ensemble subselection technique first proposed by Dobrynin et al. (2018) can be applied to large-ensemble predictions of small-scale climate extremes. Using September $T_{70}$ and November $Z_{500}$ anomalies as predictors, we were able to increase the prediction skill of the MPI-ESM-LR large-ensemble decadal prediction system for winter GBSA for both deterministic and probabilistic predictions over a hindcast period of 58 winters. Compared to the inherently low prediction skill of the full ensemble, the subselection adds value to the seasonal predictability of GBSA by improving the ACC from 0.28 to 0.64, RMSE from 0.88 to 0.70, and BSS for high storm activity against climatology from 0.03 to 0.28. The sensitivity analysis showed that the improvement of skill metrics depends on the size of the subselection and on the combination of predictors. We also showed that the skill gain can be explained through physical mechanisms, as the subselected ensemble also displays a higher ACC for deterministic predictions of winter-mean $U_{200}$ over the German Bight, as well as for the meridional gradient of MSLP and $Z_{500}$ over north-central Europe, all of which are closely related to the European winter storm climate.

### Data availability

ERA5 reanalysis products that were used to support this study are available from the Copernicus Data Store at https://doi.org/10.24381/cds.6860a573 (Hersbach et al., 2019). The 3-Hourly German Bight MSLP output data from the decadal prediction system are available at http://hdl.handle.net/21.14106/04bc4cb2c0871f37433a73ee38189690955e1f90 (Krieger and Brune, 2022a). Monthly means of MSLP, $T_{70}$, $Z_{500}$, and $U_{200}$ from the decadal prediction system are in the process of being made available. Computed German Bight storm activity time series from the decadal prediction system are available at http://hdl.handle.net/21.14106/e14ca8b63ccb46f2b6c9ed56227a0ac097392d0d (Krieger and Brune, 2022b).

### Author contribution

DK, JB, and RW designed the presented study. SB provided model data from the MPI-ESM hindcast experiments. DK analyzed the data and created the figures. All authors discussed the results. DK took the lead in writing the manuscript with input from all co-authors.

### Competing interests

The authors declare that they have no conflict of interest.

### Acknowledgements

This work has been developed in the project WAKOS – Wasser an den Küsten Ostfrieslands. WAKOS is financed with funding provided by the German Federal Ministry of Education and Research (BMBF; Förderkennzeichen 01LR2003A). Johanna

Baehr was funded by the Deutsche Forschungsgemeinschaft (DFG, German Research Foundation) under Germany's Excellence Strategy – EXC 2037 "CLICCS – Climate, Climatic Change, and Society" – project number: 390683824, contribution to the Center for Earth System Research and Sustainability (CEN) of Universität Hamburg. Johanna Baehr and Sebastian Brune were supported by Copernicus Climate Change Service, funded by the EU, under contract C3S2-370. We thank the German Computing Center (DKRZ) for providing their computing resources.

We thank the three reviewers for their valuable comments on the manuscript, as well as Joaquim Pinto for editing the paper.

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
