# Peer review of "Improving seasonal predictions of German Bight storm activity"

_EGUsphere, 2023_

## Author Comment (AC1)

Response to Reviewer #1

We, the authors, sincerely thank Reviewer #1 for their insightful and valuable comments and suggestions on our manuscript entitled *Improving seasonal predictions of German Bight storm activity*. The comments greatly helped to improve the manuscript and to remove unclarities. In the following, we would like to address the issues raised by giving a point-by-point response and propose how we plan to incorporate the comments in our manuscript.

**Comments:**

**1** The paper lacks some basic details and references on, among others, storm activity and its variability, as well as decadal prediction systems (see for example your own introduction in Krieger et al., 2022). I understand that you are trying to tell a different story than in Krieger et al. (2022), but you should not expect the reader to know your previous publication.

**Response:** We apologize for leaving out important details on storm activity and the decadal prediction system. We agree with the reviewer in seeing the need for a more independent introduction of these concepts for readers not familiar with Krieger et al. (2022) without re-telling the same storyline. We will improve the respective sections that introduce storm activity and the prediction system and add more background information.

**2** There are several relevant publications from the MetOffice colleagues on seasonal predictions, for example Athanasiadis et al. (2017), Scaife et al. (2014), or Scaife et al. (2016). Please include and discuss some of them in your paper.

**Response:** We thank the reviewer for bringing up the aforementioned publications. We will enhance the introduction and discussion and include more studies on seasonal predictability.

**3** Can you elaborate a bit why you specifically use T70 and Z500 as predictors for GBSA? Is there any reference for this choice?

**Response:** We are sorry for leaving the reasoning for this specific selection unclear. The choice of stratospheric temperature and extratropical geopotential height is based on the proposed links between the European winter climate and the QBO state, as well as the Rossby wave pattern. The choice of month and vertical level stems from a systematic lead-lag correlation test at different vertical levels with winter GBSA, which, for reasons of brevity, is not shown in the manuscript. The two predictors T70 and Z500 emerged from this analysis as the two variable-height combinations that correlated best with GBSA. It would have been similarly reasonable to use temperature or geopotential height from an adjacent height level (e.g., T50 or Z400), as the correlations between these fields and GBSA were only slightly lower. However, we decided to use those vertical levels that yielded the highest correlations. We will include a more thorough explanation of our choice in the introduction section.

**4** How do you define which ensemble members have a GBSA that is closest to the first guess?

**Response:** We thank the reviewer for this question. We select the members based on the absolute difference between the GBSA value of the respective member and the state of each predictor. We choose the *n* members that exhibit the lowest absolute difference to the first guess. We will add this explanation to the description of our methodology.

**5** Based on line 170, it looks like the ensemble has 80 members, but you only use 64 of them. Please explain this.

**Response:** The reviewer is correct here. The original ensemble consists of 80 members, 64 of which provide MSLP output at three-hourly resolution. The remaining 16 (numbered 1 through 16) only provide daily output. For this reason, we confine our analysis to the 64 members numbered 17 through 80, following Krieger et al., 2022. We will drop the member ID numbers in line 170 and clarify the choice of 64 members in the methodology section.

**6** L268-269: Can you please elaborate on why the perfect test includes future information and why this cannot be used operationally?

**Response:** Absolutely. In the perfect test, we do not select 25 members based on their proximity to the first-guess GBSA obtained from the predictor fields in September and November, but based on their proximity to the observed winter GBSA from the period December-February. While the regular subselection allows us to make a first-guess prediction for winter at the end of November (i.e., once we know the state of November Z500), the perfect test selection needs observational information from the winter itself, which, when viewed from a pre-winter standpoint, lies in the future. This perfect selection can therefore only be carried out after the winter has passed, i.e., at the end of February. For operational purposes, where a forecast for the winter is required before the start of the winter, this is impractical. We will clarify the implications of the perfect test and its limits with regard to operational use in the respective section.

**7** Outlook: How could your approach be used in future studies?

**Response:** We thank the reviewer for this question. One example for a future study based on this methodology would be to link the statistical connection between, for instance, T70 and winter GBSA to a dynamical prediction of T70 at lead times of multiple years. By establishing skillful predictions of the QBO, which governs the state of tropical T70, one or two years ahead, and then applying our method, more skillful predictions of winter GBSA one or two years into the future may become possible. In a broader context, any climate state or extreme that can be associated to physical predictors may be more skillfully predictable with our approach. While we limit our study to T70 and Z500, other atmospheric or oceanic variables may be useful as predictors for certain climate extremes, also possibly at even longer or shorter lead times. We will expand the outlook part of our discussion section to discuss these potential applications in greater detail.

---

## Author Comment (AC2)

Response to Lisa Degenhardt (Reviewer #2)

We sincerely thank Lisa Degenhardt for her constructive and insightful comments on our manuscript *Skillful Decadal Prediction of German Bight Storm Activity*. The comments greatly helped us to improve the manuscript and clarify key points.
In the following, we will give a point-by-point response to the comments and describe how we plan to address the issues raised.

**Comments:**

**a)** I have a few questions about the methods. It does seem alright, it is more the description of the method section, where I would like to see some more details for a better understanding:

**Response:** We appreciate the questions on the methodology and will address them in the following in order to improve the understanding.

**L95** My comment doesn't fit here, but maybe in the introduction or discussion, but I realised it in this line. Why are you choosing exactly these predictors? I think it would be nice to have a bit more details why you choose those and with references and why T70 is used from September.

**Response:** We apologize for leaving out details on the reasoning behind the choice of November Z500 and September T70 as predictors. The choice of stratospheric temperature and extratropical geopotential height is based on the proposed links between the European winter climate and the QBO state, as well as the Rossby wave pattern. The choice of month and vertical level stems from a systematic lead-lag correlation test at different vertical levels with winter GBSA, which, for reasons of brevity, is not shown in the manuscript. T70 and Z500 emerged from this analysis as the two variable-height combinations with the highest correlation with GBSA. Choosing Z or T at an adjacent vertical level (e.g., T50 or Z400) would have been similarly reasonable, as they are also positively correlated with GBSA. However, we decided to go with those fields that exhibit the maximum correlation with GBSA. Similarly, T70 from August or October also yields a similar correlation pattern to September T70, but the absolute correlations are slightly lower, thus leaving September as the optimal month for predicting winter GBSA. We will include a more thorough explanation of our choice in the introduction section.

**L101-Eq.1** I am a bit confused with this paragraph. And is np=nx?

**Response:** We apologize for the confusion, the $x\_n$ in Equation 1 should read $x\_p$ to fit the description in the text.

**L117-120** I think this part is a bit confusingly written or at least I got confused. I believe it does explain what Fig. 1 is showing. Which I did understood, but maybe need some more clarity

**Response:** Again, we apologize for the confusion. The text indeed is supposed to explain the workflow shown in Fig. 1. We will rephrase this paragraph to describe the subselection process more clearly.

**b) Chapter 2.5** How can you train with ERA5 data, but use the decadal data as predictor? And what are you training for?

**Response:** We are thankful for this question. The states of the predictors T70 and Z500 are always taken from ERA5 data, both for the "training" period of 1940-1959 and the "prediction" period of 1960-2019. From the decadal prediction system data, we only take the winter GBSA

predictions of the 64 individual members, as well as the large-scale fields of winter MSLP, Z500, and U200 analyzed in Section 3.2.1. The training aims to establish a set of gridpoints or areas in which the respective predictors are positively correlated with winter GBSA before starting the first actual prediction with the 1960 model run. As we have an additional 20 years available from ERA5 and observational GBSA records, we can use these 20 years prior to the start of the first model run to "train" our predictor-based selection method into knowing which gridpoints to take into account for the calculation of the predictor state, once the first model run in 1960 is started.

**c)** I think the Introduction but especially the Discussion is missing some external references. I added a personal note at the bottom, which is just a suggestion as I worked in a similar field. But in general, I would like to see a bit more referencing to other studies.

**Response:** We appreciate this comment and the suggestion of the study below. We agree that the introduction and discussion sections require a more thorough relation to other studies in the field. We will expand these sections with several relevant studies from the field of seasonal predictions of the European winter climate and put our findings into context by comparing the results of our study to those of other papers.

**Major corrections:**

**L198** I thought initialisation is November (see abstract), which I thought is mostly around the 1st and I thought you are then predicting core winter (DJF), why now "end of November"?

**Response:** That is correct, the initialization of the model takes place at the $1^{st}$ of November. However, the predictor field of November Z500 anomalies is only available at the end of the month. Hence, we can perform the subselection no earlier than the end of November. An alternative would be to use October Z500 instead which would available at or shortly after the initialization, but the correlation between October Z500 and winter GBSA is much lower than for November Z500 and would therefore not lead to any significant improvement of winter GBSA predictability. We will remove this statement in this section and add clarification to the methods section to avoid confusion here.

**Fig. 6 & 7** I don't understand the difference between Fig. 6b, e & h and Fig. 7b, e & h. I thought the sub-selected ensemble is by the 25 closest members? So for me both would be the same, but they aren't.

**Response:** Figures 6 and 7 are based on different sets of 25 members. In Fig. 6, the 25 closest members are determined via the absolute difference between predicted winter GBSA in the respective members and observed predictor fields (September T70 and November Z500). In Fig. 7, the 25 closest members are determined via the absolute difference between predicted winter GBSA and actually observed winter GBSA. Fig. 7 shows which members would have been chosen if we already knew at the start of the model run how stormy the winter would turn out to be. We realize that our description of the perfect test lacks clarity and will thus rephrase the respective section.

**L308** I understand what a "training period" is, but when I reached this part of the paper, I realised that I don't know why you needed a training period at all.

**Response:** We are thankful for raising this point. In order to determine the area-averaged state of the physical predictors in each year (T70, Z500), we need to specify the areas that we include in or exclude from the calculation of said state. We do this by correlating the predictor time series at every gridpoint with GBSA and only choosing those gridpoints that correlate significantly over

a time period from 1940 to the year before the start of the model run. As the hindcast runs start in 1960, we define the remaining time from 1940-1959 as our "training period", which acts as a starting base to get a first idea of which regions to take into account for the computation of the predictor states. This is not a training period in the classical sense, like e.g. in machine learning, but more a required "lead time" to obtain an initial view of relevant regions for GBSA prediction. We will clarify the respective section in the manuscript to make this point clearer.

**L315** I could have missed it in the paper, but where is this result coming from?

**Response:** This result is not explicitly shown in the paper. The idea of selecting those members that match the observed November Z500 was rather meant to serve as an outlook towards possible future studies that might make use of this technique. In our case, we did not find any skill gain in selecting the members this way, and therefore did not dedicate an entire section or figure to it. We realize that we should clearly differentiate between what we see as a potential outlook and what we already tested, and emphasize that in the discussion section accordingly.

**Minor corrections/suggestions:**

**L123** you only have 2 predictors, so you can say "both" instead of "multiple", right?

**Response:** That is correct, we will change "multiple" to "both".

**L134** I found this sentence a bit unnecessarily confusing written. I do understand it and it is right, but maybe it is easier to say: "ACC values of 1 indicate a perfect correlation, 0 no correlation, and -1 a perfect anticorrelation."

**Response:** We thank the reviewer for this suggestion. We will rephrase the sentence to make it less confusing.

**L152** Shouldn't $F_i$ be $O_i$?

**Response:** Here, $F_i$ refers to the probability of occurrence obtained from the prediction system. $O_i$, which we use for the observed state, is described further down in the paragraph.

**L170** What are the 17-80? Are these IDs of members? If yes, why not using the first 16?

**Response:** Yes, these are the sequential numbers of the ensemble members. For the first 16 members of the prediction system, only daily MSLP output is available, while the remaining 64 members (17 through 80) provide three-hourly MSLP output. To calculate storm activity from this output, a high temporal resolution is required and we thus decided to disregard the first 16 members. Doing so, we also keep the study consistent with the storm activity calculations in Krieger et al., 2022. We will drop the member ID numbers from the manuscript and clarify this in the methods section.

**Fig 2&3** Is there a specific reason why the dots have a white inside? The difference between white and black makes the dots quite blurry on my screen. Maybe test to make them fully black. Also in Fig. 2, I believe there is a significant area in the positive bloop over eastern Russia, maybe you could increase the density of the dots to make at least one or two dots visible there?

**Response:** We thank the reviewer for making us aware of this graphical issue. While we ourselves do not see any white in the dotted significance patterns with our PDF reader, we will try and use a

different, denser stippling pattern to avoid such visual artifacts in the future. We would greatly appreciate further feedback on whether the issue persists with the denser stippling patterns.

**L213/214** I would add here that the first sentence is for the measure correlations. Even the 25 members is right for all measures at the end. "The optimal sample size is found at 25 members per predictor for correlations (r = 0.64)."

**Response:** We agree with the reviewer and will reword the sentence.

**L219** What is Z500,sep now? I thought T70 is used from September.

**Response:**  That is correct, we apologize for this mistake. Here, it should just read Z500 without the "sep". We will change this line accordingly.

**L221** Maybe add an "individually" to make clear that you now talking about the sensitivity of each predictor alone.

**Response:** We agree with this suggestion and will add "individually" to the sentence.

**L265** Is "perfect test" and "perfect ensemble" here the same? I think I would stick with one, maybe perfect ensemble?!

**Response:** We thank the reviewer for pointing out the inconsistent wording. We intended to use perfect test for the procedure of choosing the closest members to the observed GBSA, and perfect ensemble for those members themselves. We realize that this might be confusing, and will rephrase the section to explain our definition of both terms, and to avoid switching between test and ensemble too frequently.

**L281** I believe "stark" is supposed to be "strong"

**Response:** We agree and will change "stark" to "strong".

**Personal Point:**

I did a similar skill study to seasonal predictions of European (wind-)storms and what could improve their skill. There are some publications available about that, other seasonal predictability and their influencing factors like atmospheric drivers. It would be nice to see a bit more of these studies in either the Introduction or Discussion. No need to use mine, but as we are both looking for storm activity over Europe, I wanted to mention it at least: Degenhardt, L., Leckebusch, G.C. & Scaife, A.A. Large-scale circulation patterns and their influence on European winter windstorm predictions. Clim Dyn 60, 3597–3611 (2023). https://doi.org/10.1007/s00382-022-06455-2

**Response:** We thank the reviewer for pointing us to this study. We will include the study in the introduction and discussion, as it is very relevant for this research field and should not be left out.

---

## Author Comment (AC3)

Response to Reviewer #3

We sincerely thank Reviewer #3 for their constructive and insightful comments on our manuscript *Improving seasonal predictions of German Bight storm activity*. The comments greatly helped us to improve the manuscript and clarify key points.
In the following, we will give a point-by-point response to the reviewer's comments and describe how we plan to address the issues raised.

**Major comments:**

**1** The study should provide a bit more historical context of their subsampling idea in the introduction. Also, the Discussion section does not refer to any other studies.
Historical context: Conceptionally, the approach returns to an old idea of "analogue forecasting" of future states that can be constrained based on their similarity to meaningful predictors of the preceding observed state (cf. Lorenz 1969; Barnett et al. 1978). As demonstrated earlier for weather prediction (van den Dool 1994) or weather field reconstructions (Schenk & Zorita 2012), the skill will depend on various choices that Krieger et al. test here for a large-ensemble prediction system. In particular, the skill of the subsampling approach will also depend on the number of spatial degrees of freedom of predictor and target variables.

**Response:** We appreciate the comments and input on the historical context of subsampling and analogue forecasting. We will expand the introduction to properly present the idea of subsampling and its origin. We will also discuss more studies in the discussion section.

**2** Statistical context: In summary, the authors make largely appropriate efforts to provide robust statistical significance testing for both, identification of predictors as well as the resulting prediction skill. While the authors consider temporal autocorrelation using block-bootstrapping, I have some concerns that their locally significant results could be randomly significant owing to the potentially very low number of spatial degrees of freedom and hence very large spatial autocorrelation of fields like T70 and Z500. A quick field significance test is suggested below (cf. Livezey & Chen, 1983; Wilks, 2006) which would otherwise not change the results of the final prediction skill in this study. In the historical context, this study is even more remarkable as a quite good skill is achieved with using spatially rather homogeneous predictors.

**Response:** We appreciate the feedback on the statistical significance testing. We see the need to include a global significance test that checks the proportion of local tests which were erroneously considered significant. We decided to perform a global test that controls the false discovery rate (FDR) as it is considered one of the most powerful tools to check global field significance (e.g. in Wilks, 2006). We find that by controlling for the FDR at a level of 0.05, 21% of gridpoints for T70 and 82% for Z500 that were previously considered significant (7% and 13% of all gridpoints, respectively) are now insignificant. However, those areas that were deemed most relevant for the prediction of GBSA, namely the tropics for T70 and the extratropical Rossby wave train for Z500 are still significant globally. Thus, we decided to keep calculating the predictors from locally significant gridpoints. We will add this test to the methodology section and refer to it wherever necessary. We updated Figures 2 and 3 in the manuscript (see Figs. R1 and R2 below) to now display both locally-only and locally-and-globally significant gridpoints.

[Figure]

**Fig. R1** Gridpoint-wise correlation coefficients between global T70 anomalies in ERA5 and observed winter (DJF) German Bight storm activity. Period 1940–2017 for temperature anomalies, 1940/41–2017/18 for storm activity. Hatching indicates local statistical significance (p ≤ 0.05) determined through 1000-fold bootstrapping. Stippling indicates additional global (field) significance by controlling for the FDR at a level of 0.05.

[Figure]

**Fig. R2** As Fig. R1, but for Z500 instead.

**Minor comments:**

**L28** You directly jump here to the concept of using specific physical predictors to aid in reducing the spread of model predictions and increase prediction skill. You later use this idea in this study to improve subsampling of certain ensemble members that are most similar to an initial predictor state. It might be worthwhile to briefly mention here some historical context mentioned above that this idea is very similar to analogue forecasting attempts already in 1970s to predict future weather (Lorenz 1969) or short-term climate fluctuations (Barnett et al. 1978) based on analogues that proceed from the present state to estimate future states. Also, the idea to predict unknown full field states based on incomplete low-order predictors via analogues was successfully applied in reconstructions (e.g., Schenk & Zorita 2012). Interestingly, in that study the skill improvement was tested in a very similar way as done here regarding the dependency on the number of predictors, use of multivariate predictors and benchmarking with an idealized model-dependent prediction skill.

**Response:** We greatly appreciate the input on the close relation between predictor-based predictions and analogue forecasting. As mentioned in our response to the 1$^{st}$ major comment, we will expand the introduction to properly introduce analogue forecasting and the idea of subsampling.

**L97-99** It is a bit unclear how statistical significance is derived here. Fields of T70 and Z500 tend to have a very large spatial autocorrelation (low spatial degree of freedom). It is quite likely that far more than 5% with easily up to more than 20% of locally significant grid cells could be randomly significant globally. To be sure, you could test the global significance by estimating 1000x correlations from bootstrapping of these fields and its correlation with ERA5 fields to evaluate how many grid cells are randomly significant (Wilks 2006). If your T70 and Z500 predictor fields vs. ERA5 fields yield more locally significant grid cells than the randomly significant correlations, you can claim to use globally or regionally significant predictor fields. It should be noted that even if this test fails, predictor fields may still provide predictive skill if the locally significant areas are physically meaningful (e.g., linked to Rossby waves, NAO etc.). It looks like this is the case in your study. You could add a plot of the global field with random correlations and locally significant areas and provide the test quantity of "overall % of n significant grids x 100 / N total grids" which provides more context to Fig. 2 and 3 in the main text. Based on the constraints (line 113), global in this study could also be regionally 30-90°N here. Based on Fig. 2, this area may not be overall significant but appears to provide coherent regions of locally significant correlations. Fig. 3 might show a higher fraction of significant correlations due to wave propagation (N=4 areas) plus Arctic, total N=5 areas of predictive skill but perhaps only N=2 independent predictors?

**Response:** We agree with the reviewer that the manuscript is missing a global significance test. We will add a field test based which controls for the FDR at a level of 0.05, and add the results to the correlation maps in Figs. 2 and 3, as well as the corresponding text.

**L102-106** Here, the randomly significant issue becomes obvious from using a gridpoint-wise testing with a 95% local confidence level. The selection process is correct locally but may not provide regionally or globally significant field results regarding the full fields with low spatial degrees of freedom. I would not change this procedure here but just add a small test described above to include a sentence whether you use regionally or globally significant predictor fields or only locally significant results. Despite using fields, your predictor would then be local.

**Response:** We appreciate this insight. As mentioned in our response to the 2$^{nd}$ major comment, we will include global significance testing and compare the global significant fields to the locally

significant gridpoints. We will keep the predictor local, but explicitly add this to our methodology description.

**L186-191** Perhaps mention here that the four significant areas in the northern extra-tropics represent the Rossby wave propagation in addition to teleconnections (Arctic Oscillation-like?) with the Arctic versus (sub-)tropical areas of Sahel and Indian Ocean. It is quite nice to see that these physically meaningful areas show up also statistically.

**Response:** We thank the reviewer for suggesting to add the Rossby wave pattern and the AO here. We expanded the paragraph to include these large-scale patterns.

**L223-224** "purely coincidental". Not a coincidence at all. There is a direct relationship between the correlation coefficient and the RMSE, i.e. when standardized observations and predictions are used as RMSE inputs (hence bias = 0). This means that the RMSE is a measure of the unexplained variation, which is inversely proportional to the explained variation, which is the square of the correlation coefficient (as can be seen in Fig. 4). Therefore, it is not purely coincidental that for both predictors the optimal sample sizes for RMSE and correlation are equal, but a consequence of the mathematical relationship between these two statistics. Please replace the sentence with the opposite statement.

**Response:** We apologize for the incorrect statement. The RMSE and correlation coefficient are indeed related. We will correct the mistake and replace the sentence, also referring to a description of this relationship in Barnston (1992).

**Figure 5** Very good illustration and impressive result.

**Response:** We thank the reviewer.

**L265-267** I generally like that test regarding the question what the best selection of the 25 members would be knowing the observed state. Here, you could've gone even further by evaluating the single best member per year out of 64 members relative to ERA5 for 1960-2017/18. That would be a prediction-system-specific optimum of the ensemble initialised in late autumn for DJF which could be compared to your "almost perfect test".

**Response:** We thank the reviewer for the suggestion to test the large-scale prediction skill of the single best member per year. The results are shown in Fig. R3 below. For the single best member, we identify an increase in ACC for MSLP in west of Norway and west of Iberia, as well as a slight decrease over the North Sea. For Z500, the ACC is increased northeast of Iceland, but reduced over the North Sea and the majority of Europe. For U200, we find a slight increase over the North Sea, but a stronger decrease from the Central Atlantic to the Alps, and also from Greenland into northern Scandinavia. Compared to Fig. 7 in the manuscript (i.e, the perfect test for 25 members), the ACC change is overall more negative, but similar features emerge, such as the increase of U200 over the North Sea, and the tripole-like structure of ACC changes for Z500 and MSLP. We speculate that a single member might not be sufficient to predict the same large-scale patterns in the same region correctly every year, which could then lead to lower correlation gains overall than for a 25-member ensemble mean.

[Figure]

**Fig. R3** Anomaly correlation coefficients (ACC) for ensemble mean predictions of the full 64-member ensemble (left column), the 1-perfect-member-subselection (middle column), and the change in ACC between the full ensemble and the perfect member (right column) for winter-mean (DJF) MSLP anomalies (first row), 500 hPa geopotential height anomalies (Z500, second row), and 200 hPa zonal wind anomalies (U200, third row). Winter-mean anomalies are calculated by averaging monthly anomalies from December, January, and February. Period 1960/61–2017/18. Stippling indicates statistical significance (p ≤ 0.05) determined through 1000-fold bootstrapping

**L282** Agree. I guess here you could mention the potential for machine learning methods.

**Response:** We thank the reviewer for this suggestion and will add a remark possible involvement of machine learning techniques.

**Figure 8** How much do these composites differ from a first-year composite? Could the strong Antarctic difference be caused by a long-term trend in the model runs over time rather than highlighting differences from the composites?

**Response:** We appreciate this comment. The first-year composite (see Fig. R4 below) displays a similar behavior as the composite of all lead years (Fig. 8 in the manuscript), with a strong signal in the Southern Ocean and Antarctic regions and composite close to 0 K in the tropics, this time even slightly negative instead of slightly positive.

[Figure]

**Fig. R2** Composite mean T70 of 100 model years with the highest subsequent DJF GBSA minus composite mean T70 of 100 model years with the lowest subsequent DJF GBSA in MPI-ESM-LR decadal hindcast runs. Data are taken from all initializations, all members, but only the first lead year after the initialization.

**L299** The whole discussion section does not make any attempts to put results into context with other seasonal prediction studies (e.g., Kruschke et al. 2014; 2016 and many others). I see that some relevant studies were briefly discussed in Krieger et al. (2022) but not here. I suggest adding a paragraph or several sentences throughout chapter 4 where similarities and differences to other studies are discussed.

**Response:** We thank the reviewer for this comment and apologize for not mentioning more studies from the research field in the discussion section. We will expand both the introduction and discussion with relevant studies and discuss our results by putting them into context through comparison with the findings of similar studies.

**L300-303** Although you're using a decadal prediction system, does that really differ from using a seasonal prediction system in your specific case? The initialisation in November to predict DJF is pretty much what a seasonal prediction system would do.

**Response:** We thank the reviewer for voicing this concern. In our case, that is, using the MPI-ESM prediction systems, the decadal and seasonal systems differ mostly in ensemble size and resolution. The decadal prediction system consists of 80 members, 64 of which provide 3-hourly resolution, while the seasonal system only contains 30 members at 6-hourly resolution. Secondly, the decadal system runs at low spatial resolution (T63 grid, 1.8 degree grid spacing), while the seasonal system runs at high spatial resolution (T127 grid, 0.9 degree grid spacing). The higher resolution in the seasonal system is also present in the number of vertical layers, which results in the seasonal system being able to maintain a QBO signal, whereas the lower vertical resolution in the decadal system is unable to properly maintain the QBO in the stratosphere. There are also minor differences in the nudging data of the respective assimilation runs, where in the seasonal system ERA5 is used for all hindcasts, while the decadal system uses ERA-Interim until 2015 and ERA5 from 2016 onwards. Other than that, the two systems are very similar.

**L305** Most likely because the annual GBSA is dominated by the variation in winter (high correlation of high annual percentiles with high winter percentiles, i.e. same tail values)?

**Response:** This is likely the case, yes. Winter contributes most to the annual storm activity metric as the highest wind speeds occur during the winter months and, therefore, the upper percentiles of winter and the entire year show a high correlation. We will add an explanatory sentence to the discussion.

**L308** "by two decades"

**Response:** We thank the reviewer for this correction.

**L321** Regarding NAO, perhaps AO would be more appropriate as mentioned above?

**Response:** We appreciate this suggestion. We correlated the tropical September T70 predictor time series and the DJF Arctic Oscillation Index for the time period 1960/61-2017/18 and found a correlation coefficient of 0.11. To put this into perspective, the correlation between T70 and the DJF NAO index for the same time period is 0.27, while the correlation between T70 and DJF GBSA is 0.49. We will mention AO in this section alongside the NAO, as we already added the AO to the analysis of the correlation maps and see the need to refer back to it here.

**References:**

Barnston, A. G. (1992): Correspondence among the Correlation, RMSE, and Heidke Forecast Verification Measures; Refinement of the Heidke Score. *Weather and* Forecasting, 7 (4). DOI: 10.1175/1520-0434(1992)007<0699:CATCRA>2.0.CO;2

---

## Author Response (AR2)

Dear Joaquim Pinto,

We, the authors, would like to thank you for the opportunity to submit the finalized version of our manuscript entitled "Improving seasonal predictions of German Bight storm activity" to *Natural Hazards and Earth System Sciences*.

We are glad and honored to hear that our manuscript was selected as a highlight paper, and would like to thank the reviewers again for their invaluable comments and input during the review process. We have incorporated all remaining corrections into the finalized version of our manuscript and look forward to hearing from you regarding the next steps of the publication process.

Sincerely,

Daniel Krieger

Corresponding Author